# PVB Nanocomposites as Energy Directors in Ultrasonic Welding of Epoxy Composites

Fabrizia Cilento [1], Alessio Bassano [2], Luigi Sorrentino [1], Alfonso Martone [1,*], Michele Giordano [1] and Barbara Palmieri [1]

1   IPCB, Institute of Polymers, Composite and Biomaterials, National Research Council of Italy, P.le E. Fermi, 1, 80055 Portici, Italy
2   Leonardo Electronics, Defense Systems Business Unit, Via Valdilocchi 15, 19126 La Spezia, Italy
*   Correspondence: alfonso.martone@cnr.it or alfonso.martore@cnr.it; Tel.: +39-08-1775-8816

**Abstract:** Ultrasonic welding (UW) is a well-established technique for joining thermoplastic composites and has recently been utilized in the aerospace and automotive industries. In the case of thermoset composites (TSCs), a polymer-based material placed at the welding interface called an energy director (ED) is required. The choice of the coupling layer material is linked to several requirements, such as processing temperature, high adhesion to the thermoset composites (TSCs) adherend and mechanical strength of the resulting welded joints. In this work, the authors investigated the possibility of using Poly-vinyl-butyral (PVB) reinforced with graphite nanoplatelets (GNPs) as a coupling layer in the UW of TSC adherents. The effect of GNPs aspect ratio and content on the weldability of carbon fiber-reinforced plastics (CFRP) has been investigated. PVB/GNPs nanocomposites with different filler contents (from 0.5 wt% to 2 wt%) and different aspect ratios (100 and 2100) have been fabricated. The influence of the viscoelastic properties of the flat EDs on weldability has been assessed. Finally, an improvement of lap shear strength (LSS) of 80% was found for nanocomposites with 0.5 wt% of high-aspect-ratio GNPs with respect to neat PVB. The use of high damping nanocomposites as coupling materials for TSCs paves the way for a new generation of EDs in UW.

**Keywords:** nanocomposites; damping; ultrasonic welding; graphite nanoplatelets

## 1. Introduction

In the case of thermoset composites (TSCs), the most used techniques for joining of TSCs are mechanical fastening and adhesive bonding, or a combination of the two [1]. Both techniques have disadvantages: in the first case, the stress concentrations due to the drilling of holes results weaken the thin composite structure and reduce the lightweight potential [2,3]; in the second case, rigorous surface preparation, high curing temperatures and pressures and long curing times are required to preserve the long-term durability of bonded composite joints [4].

To overcome these drawbacks, the development of novel joining techniques is required. Welding is a highly efficient process for joining composites, being capable of producing joints in relatively short cycle times, characterized by equivalent or better performance than adhesively bonded or mechanically fastened joints.

Specifically, ultrasonic welding (UW) is a well-established technique for joining composites and has recently been utilized in the aerospace and automotive industries [5,6]. However, it requires the melting or softening ability of the polymer with increasing temperature, being applicable only in joining fiber-reinforced thermoplastic composites (TPCs) [7,8]. To weld TSCs, the use of polymer-based films placed at the welding interface is required to promote frictional and viscoelastic heating [9]. These coupling films, called energy directors (EDs), must satisfy various requirements such as processing temperature, high adhesion to the TSC adherend, and mechanical and environmental performance of the resulting welded joint [10].

The PEI interlayer is widely studied for assembling TPCs, such as CF/PEEK composites, as a way to prevent deconsolidation and degradation due to overheating, thanks to the low melting temperature compared to the TPC of the adherents and the ability to easily flow at the interfacial zone due to its low viscosity [11,12].

Recent studies demonstrated the possibility of also using ultrasonic welding for welding thermoset composites. PEI coupling layers have been widely employed to weld CF/epoxy and CF/PEEK adherents, showing a promising average lap shear strength (LSS) of 28 MPa [13]. Few studies include the use of other thermoplastic polymers, such as PEEK [14] or PA6 [15], to weld CF composites with the same substrate's polymeric matrices. A hybrid system has been investigated by Lionetto et al. [16], who used flat PVB films as EDs in the ultrasonic welding of thermosetting matrix composites, co-cured on the top surface of CFRP laminate, showing an LSS of 27 MPa.

In contrast to other welding processes, in UW, heat is generated by the internal damping of an applied ultrasonic vibration. During ultrasonic welding, the parts are subjected to a longitudinal mechanical vibration with a high frequency (typically 10–40 kHz). The mechanical vibration causes a standing wave in the welding parts in the form of heat [17]. The mechanism of heat generation is via viscoelastic dissipation due to the cyclic deformation of the plastic. In a purely elastic material, the driving force causes deformation rates in the part that is in phase with the force, so there is no energy dissipation. The deformation rate in a purely viscous material is proportional to the applied force but has a phase difference of 90° dissipating the energy. Thus, in UW, the driving dynamic welding force depends on the material properties and shape of the product when the ED heats, melts, and flows to fill the interface joining the parts. With increasing temperature, the damping factor grows and the increasing vibrational energy is converted into heat by hysteresis losses [18]. The transformation of mechanical vibration into heat depends on the mechanical and thermal properties of the material: the damping must reach a certain level to cause high hysteresis losses, which are responsible for the elevation of the temperature. On the other hand, the material must be stiff enough to successfully transmit the vibrational energy from the horn to the joint interface. The polymer itself may not reach these mechanical requirements.

Nanomaterials can effectively increase the mechanical performances of polymers both in terms of elastic modulus and damping, thanks to the energy dissipation that occurs at the interface with the matrix [19]. Nanoparticles with a lamellar (2D) structure, such as graphene and its analogues (graphene oxide, GO, graphene nanoplatelets, GNP, etc.), can significantly improve the damping capabilities of nanocomposites [20,21], thermal conductivity [22] and fracture toughness [23]. The nanoparticle shape factor and content can influence the mechanical behavior of nanocomposites and in particular the damping factor (tanδ) [24]. The higher the nanoplatelet aspect ratio, the higher the damping capacity, because of an enormous increase in surface area. An interfacial stick-slip mechanism between the polymer matrix and nanofillers accounts for the enhancement of damping in nanocomposites [25].

Viscoelastic and surface frictions are the leading heating systems in UW. The heat generated during the vibration is concentrated at the interface via the ED. During polymer welding, primary heat generation is due to interfacial friction, whereas secondary and main heat generation is due to viscoelastic heating [26]. Li et al. [27] investigated the effect of MWCNT-based energy director films on the UW process of glass fiber/polypropylene (GF/PP), showing their potential for damage detection through real-time electrical resistance changes at the interface.

In this work, the authors investigated the possibility of using Poly-vinyl-butyral (PVB) reinforced with graphite nanoplatelets (GNPs) as a coupling layer in the UW of TSC adherents. The effect of GNPs' weight content and flake dimensions on the temperature-dependent dynamic characteristics of nanocomposites and their influence on joint quality have been assessed. Specifically, nanocomposites with GNPs content varying from 0.5 wt% to 2.0 wt% and different aspect ratios have been fabricated using a lab mixer. Both the

viscoelastic behavior and morphology of the nanocomposites have been investigated. Finally, the interlaminar shear properties, through the lap shear strength and short beam strength of UW joints, have been investigated.

## 2. Materials and Methods

### 2.1. Materials and Nanocomposites Preparation

Poly-vinyl-butyral (PVB) (Mowital B60H) is a tough plastic resin commonly used for bonding. The presence of hydrophilic vinyl alcohol units and the hydrophobic vinyl butyral units confer both high adhesion to inorganic material and elasticity and toughness.

Two-dimensional graphite nanoplatelets (GNPs) are multilayers of graphene obtained by the exfoliation of graphite. Nanoplatelets with different aspect ratios were selected in this study: MICRO850, purchased by Asbury (5 μm lateral size, 50 nm thickness, 14 m$^2$/g SSA, aspect ratio 100) and G2NAN, kindly supplied by NANESA (30 μm lateral size, 14 nm thickness, 30 m$^2$/g SSA, aspect ratio 2100).

Nanocomposites filled with selected GNPs were produced following a mixing and compression molding process. Firstly, the polymer and the nanoplatelets were mixed using a Thermo Haake Rheomix for 10 min at a temperature of 150 °C. Then, films of 200 μm thickness were fabricated using a hot platen press at 140 °C and 200 bar for 20 min.

Coupons with different filler content were fabricated, varying the filler concentration and aspect ratio (Table 1). Nanocomposites made with MICRO850 are specified as LAR (lower aspect ratio) while nanocomposites made by G2NAN are specified as HAR (higher aspect ratio). A bare PVB film was produced as a reference material for comparing composite performances.

**Table 1.** List of fabricated nanocomposites.

| Sample | Filler | Nominal Filler Content [wt%] |
|--------|--------|------------------------------|
| Ref | Neat | - |
| 1-HAR | G2Nan | 0.5 |
| 2-HAR | G2Nan | 1.0 |
| 3-HAR | G2Nan | 1.5 |
| 4-HAR | G2Nan | 2.0 |
| 5-LAR | MICRO850 | 0.5 |
| 6-LAR | MICRO850 | 2.0 |

### 2.2. Composites Manufacturing and Welding Procedure

A carbon fabric/epoxy prepreg with a fiber volume content of 58% was used.

Laminates with a nominal thickness of 2 mm were fabricated by stacking 8 plies of prepregs with the following stacking (Figure 1). The CF/epoxy stack was cured in an autoclave at 120 °C for 2 h and 3 bar with a vacuum bag.

Specimens cut from the laminates were ultrasonically welded using the PVB/GNP film as the coupling layer and ED. Ultrasonic welding of individual coupons was performed using a Rinco Dynamic 3000 ultrasonic welder (Figure 1). The flat ED was placed between two CF/epoxy substrates and welded by the following process parameters: welding force of 500 N, peak-to-peak amplitude of 36 μm, welding time of 1000 ms and maximum welding energy of 500 Ws.

### 2.3. Experimental Characterization

Thermogravimetric analysis (TGA) (TA Instruments Q500) was conducted to evaluate the real filler/matrix composition of the films, according to ASTM E1131. Measurements were performed in an inert atmosphere, using nitrogen gas, with a temperature ramp of 10 °C/min from room temperature to 700 °C. The weight loss was evaluated at 500 °C.

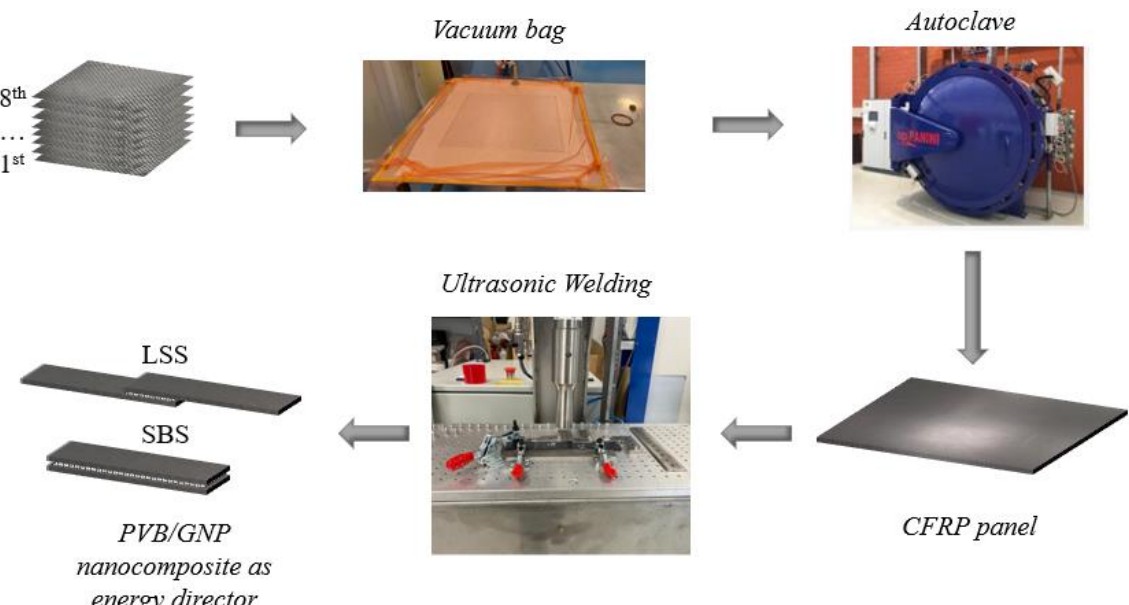

**Figure 1.** Fabrication process of CF/epoxy laminates and set-up of the ultrasonic welding process.

The thermal properties of the PVB/GNPs films were investigated by differential scanning calorimetry (DSC) using the DSC Discovery instrument. Each specimen was heated and cooled twice from 0 to 300 °C at a rate of 10 °C/min and the glass transition temperature ($T_g$) values were extracted from the DSC curves, according to ASTM D3418.

Dynamic mechanical analysis (DMA) was employed to measure the loss modulus and assess the viscous behavior of the PVB/GNPs nanocomposites. A TA Instruments Q800 DMA equipped with a single cantilever clamp was used to perform a temperature sweep from 30 to 120 °C at a heating rate of 3 °C/min and a frequency of 1 Hz and considering an initial amplitude of 20 μm. Data are elaborated according to the ASTM D790 standard for the flexural behavior of composites [28].

An optical microscope Olympus BX 51M, equipped with Linkam THM600 hot stage, was employed to assess the morphology of the material. Thin PVB/GNP films of 200 μm thickness were prepared and the quality of the dispersion of the nanoparticles were assessed in transmission mode.

The coupons were cut from the cured CF/epoxy plates to the dimensions of 36 × 12 mm for the Interlaminar Shear Strength (ILSS) tests (Figure 1) and 25 × 100 mm for the Lap Shear Strength (LSS) tests (Figure 2).

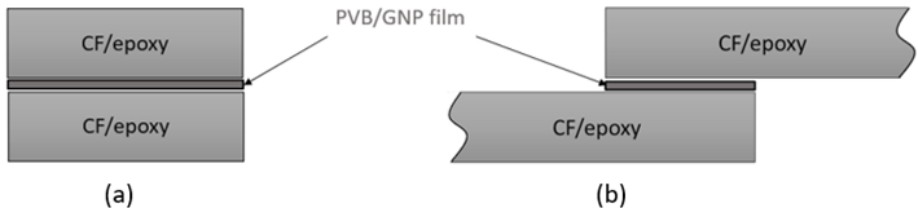

**Figure 2.** Schematic representation of the CF/Epoxy/PVB joints in (**a**) ILSS and (**b**) LSS configurations.

ILSS tests were performed according to ASTM D2344 standard to evaluate the interfacial strength of the ultrasonically welded joints; the bonding characteristics of the ultrasonic welded joint were evaluated by LSS tests, according to ASTM D5868 standard. The tests were carried out with a 50 kN load cell in an Instron 68TM-50 Mechanical tester.

The interlaminar shear strength (ILSS, MPa) was calculated by dividing the peak recorded reaction force ($F_{max}$, N) by the area of the cross-section ($A_{ILSS}$, mm$^2$), equal to $40 \times 12$ mm$^2$, as shown in Equation (1).

$$\text{ILSS} = \frac{3}{4} \cdot \frac{F_{max}}{A_{ILSS}} \tag{1}$$

The lap shear strength (LSS, MPa) of the joints was calculated as the maximum load ($F_{max}$, N) divided by the total overlap area ($A_{LSS}$, mm$^2$). The total overlap area was about $25 \times 25$ mm$^2$, Equation (2).

$$\text{LSS} = \frac{F_{max}}{A_{LSS}} \tag{2}$$

A minimum of three specimens were tested per type of PVB/GNP coupling layer used in this study.

## 3. Results

### 3.1. PVB Nanocomposites

The results of thermal analyses conducted on nanocomposites are reported in Table 2. A good agreement between the nominal filler content and the actual filler content was found for all selected samples. The actual filler content is calculated according to Equation (3).

$$w_{f,actual} = \frac{R_{NC} - R_m}{1 - R_m} \% \tag{3}$$

where $R_{NC}$ and $R_m$ are the residues at 500 °C of the TGA conducted on the nanocomposites and the PVB matrix, respectively.

**Table 2.** Results of thermal analyses conducted on PVB/GNPs nanocomposites.

| Sample | Actual Filler Content [wt%] | $T_g$ [°C] |
|---|---|---|
| 1–HAR | $0.50 \pm 0.05$ | $72 \pm 2$ |
| 2–HAR | $1.00 \pm 0.03$ | $73 \pm 1$ |
| 3-HAR | $1.50 \pm 0.08$ | $73 \pm 2$ |
| 4-HAR | $2.10 \pm 0.15$ | $73 \pm 1$ |
| 5-LAR | $0.50 \pm 0.06$ | $72 \pm 3$ |
| 6-LAR | $2.20 \pm 0.10$ | $72 \pm 2$ |

In Table 2, the values of the glass transition temperature ($T_g$) are listed, indicating that the $T_g$ of the polymer is not affected by the inclusions. The average glass transition according to DSC experiments is 73 °C for all specimens, and no variation in the aspect ratio or content of GNP was found.

Micrographs show the dispersion state in the case of MICRO850 and G2Nan. In t case of G2NAN nanocomposites (HAR), the distribution of nanoparticles is mostly uniform for all concentrations from 0.5 wt% to 2.0 wt%, with a slight level of agglomeration at high GNP content (Figure 3b,c). However, MICRO850 nanocomposites (LAR) showed a better level of homogeneity with respect to G2Nan nanocomposites (HAR) and the absence of clusters, even at high filler content (2 wt%).

The geometry of nanoparticles affects the state of dispersion since the higher the aspect ratio, the lower the statistical percolation threshold. The percolative threshold is a critical content above which the nanoparticles should not be isolated and start to agglomerate [29–31].

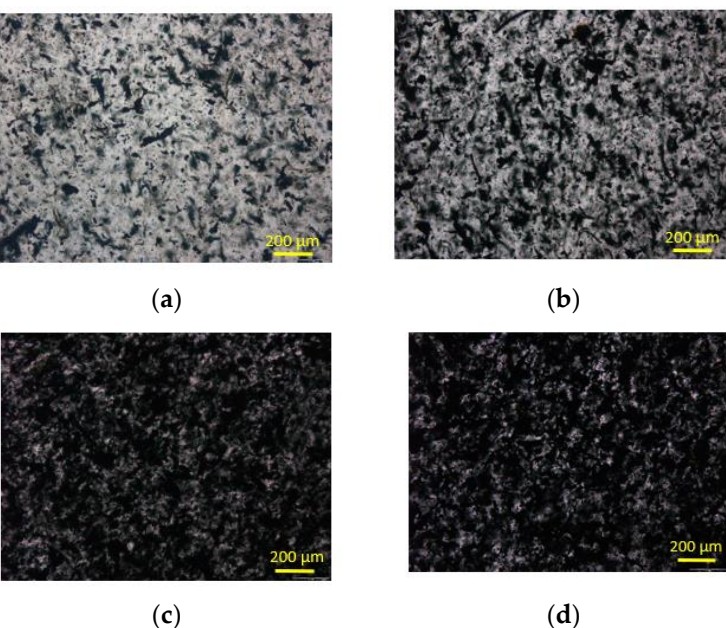

**Figure 3.** Micrograph of PVB reinforced with G2Nan at 0.5 wt% (**a**), 1 wt% (**b**), 1.5 wt% (**c**) e 2 wt% (**d**).

The percolation threshold in the case of nanocomposites reinforced with the 3D random orientation of platelets depends on the nanoplatelets' aspect ratio. The higher the aspect ratio, the lower the volumetric filler fraction for which the nanoparticles agglomerate, as indicated in Equation (4) [29,32]:

$$v_f = \frac{27\pi D^2 t}{4(D + D_{IP})^3} \sim \frac{27\pi t}{4D} \tag{4}$$

where $D$ and $t$ are the nanoplatelet lateral size and thickness, respectively, and $D_{IP}$ is the interparticle distance [32].

In the case of G2Nan, the percolation threshold is around 1.9 wt%, while in the case of MICRO850, this value reaches 34 wt%.

Thanks to the small particle size and the high percolation threshold, LAR nanocomposites are still homogenous at the higher filler content (2%wt, Figure 4b), whereas HAR nanocomposites start agglomerating at 0.5 wt% (Figure 3a). Therefore, the mixing process was not effective in dispersing the nanoplatelets.

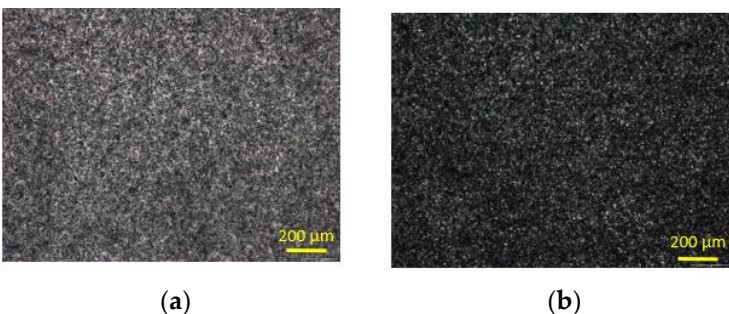

**Figure 4.** Micrograph of PVB reinforced with MICRO850 at 0.5 wt% (**a**), 2.0 wt% (**b**).

The GNPs dispersion condition affects the mechanical performance of the nanocomposite. In fact, a linear increase in the elastic modulus with filler content is found in the case of PVB loaded with MICRO850. The results of the dynamic mechanical analyses conducted on the samples, in terms of storage and tan$\delta$, are reported in Figure 5.

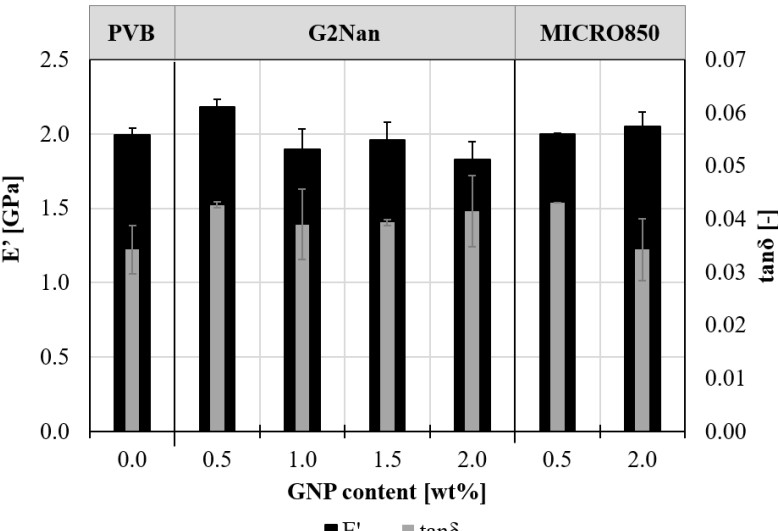

**Figure 5.** Comparison between storage modulus and damping ratio at room temperature of nanocomposites.

In the case of G2Nan composites, a maximum storage modulus of 2.3 GPa (*E′*) is found for samples with a G2Nan content of 0.5 wt%, higher than previously reported data [33], with an increase of 9% compared to unfilled PVB. For higher filler contents, a decrease in mechanical properties compared to the neat PVB is observed due to the agglomeration of GNPs.

Figure 5 reports the effect of filler content on the mechanical and dissipative properties of PVB nanocomposites. The mechanical performance depends strictly on the state of dispersion of GNP within the hosting matrix; in the HAR nanocomposites the elastic modulus slightly decreased as the agglomeration of the particles increased, while in the LAR nanocomposites the elastic modulus increased according to the filler loading since no agglomeration occurred.

In both cases, a substantial increase in tan*δ* was measured for samples at lower GNP contents (0.5 wt% of GNP) compared to the bare PVB, independently of the filler aspect ratio.

By increasing the filler percentage, the HAR nanoplatelets form local clusters since there is an abrupt increase in viscosity, which threatens the mixing quality. As a result, a reduction in stiffness and damping was observed, likely related to the poor interface between the GNPs and the matrix and the presence of agglomeration particles.

At low particle content, there was a concurrent increase in mechanical performance and dissipative behavior, while the poor dispersion at higher content worsened the stress transfer, resulting in a reduction in storage modulus. In addition, the internal slippage between particles leads to a progressive increase in energy dissipation [25].

The analysis of dissipative behavior shows that independently of the aspect ratio, the tan*δ* decreases. In HAR composites, the drop of tan*δ* is smooth since aggregates appeared but at lower filler content, while in LAR composites the tan*δ* drop is remarkable (15%) and probably related to the capability of the filler to transfer the load to the hosting matrix, promoting a stiffer behavior.

### 3.2. Mechanical Properties of Welded Joints

In UW, the energy directors need to match damping and stiffness requirements to guarantee a certain level of hysteresis losses and to successfully transmit the vibrational energy from the horn to the joint interface, respectively.

A strict correlation between the viscoelastic properties of the interlayer (PVG/GNP) and the short beam shear strength (SBS) is observed (Figure 6). The mechanical properties of the coupling layer influence the strength of the joint.

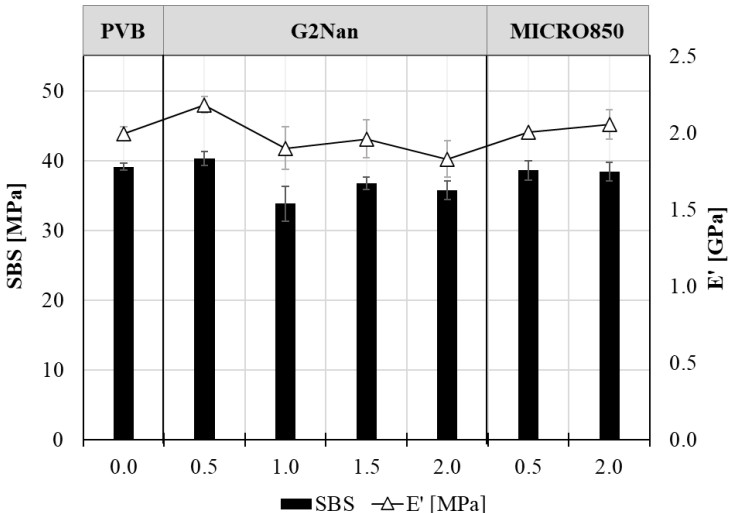

**Figure 6.** Comparison between SBS and storage modulus.

By analyzing interlaminar strength, there are two different behaviors: below a critical content (0.5 wt%), the presence of nanoparticles does not affect the strength, and the ILSS measured is very close to the ILSS achieved with bare PVB film. Above the critical content (PVB/G2nan nanocomposites), the interlaminar strength decreases according to the mechanical performance of the interlayer.

Welded joints with nanocomposites with a high elastic modulus showed a SBS compared to the neat PVB, with a maximum of 40 MPa for PVB filled with 0.5 wt% of G2Nan. At low nanoparticle content, the interlaminar strength of polymers improves, since the inclusions delay crack propagation [23]. At this stage, only nanocomposites reinforced with G2Nan have been employed for further investigation.

Compared to neat PVB, the lap shear strength increases for all HAR nanocomposites, showing a maximum in the case of 0.5 wt% filler content (Figure 7). As for mechanical properties (Figure 5), two different regimes are observed: below 0.5 wt%, the strength increases, and above 0.5 wt%, a sudden drop of strength followed by a subsequent increase is observed (Figure 7).

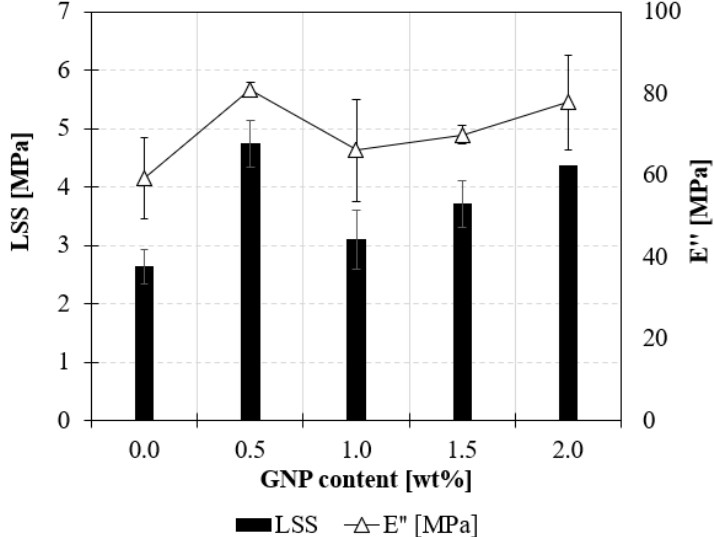

**Figure 7.** Comparison between LSS and loss modulus for nanocomposites reinforced with HAR nanoplatelets.

## 4. Discussion

The obtained results encourage the use of nanocomposites for the hybrid welding of TSCs. The majority of studies mainly refer to the use of PEI as an ED [10,14,34], and only a few authors investigated the possibility of using other thermoplastic polymers for joining CF/epoxy composites. The use of PVB represents a valid option due to the chemical compatibility with epoxy resin, as reported by Lionetto et al. [16].

In this work, a further step forward into a deep knowledge of ED behavior has been introduced based on the inclusion of graphite nanoplatelets in the PVB matrix, with the aim to improve the dissipative behavior of the EDs. The choice of PVB as a bulk material is linked to the low melting temperature (150–170 °C) compared to PEI (greater than 200 °C) as well as the excellent binding and film-forming ability, adhesion to many surfaces and good interaction with graphitic nanoparticles.

As stated in the work of Menges et al. in 1971 [35], ultrasonic weldability is affected by the loss modulus of the energy director. During welding, the energy is concentrated at the level of the energy director, which heats, melts and flows, joining the parts. With increasing temperature, the damping factor of the ED grows, and the vibrational energy conversion is converted into heat by hysteresis losses.

The increase in LSS is strictly related to the quality of the joint. A non-dimensional parameter should be introduced as a measure of the ultrasonic weldability, $\Phi$ (Equation (5)) [18].

$$\Phi = \int_{T_1}^{T_1} \frac{\rho c_p}{E'\left[\frac{\tan\delta}{2} + \frac{0.25}{\pi}\left(1 - \left(\frac{p}{p_k}\right)^{0.7}\right)\mu\right]} dT \tag{5}$$

where $\rho$ is the material density, $c_p$ is the specific heat capacity, $p/p_k$ is the ratio between the welding and static pressure, $\mu$ the friction coefficient, $T$ is the temperature and $E'$ and $\tan\delta$ the storage modulus and the damping ratio of the energy director, respectively.

The $\Phi$ parameter describes the specific energy consumption required for welding the two parts. The weldability is strongly affected by the loss modulus of the material, i.e., by the product between $E'$ and $\tan\delta$. The higher the storage modulus and the damping capacity of the ED, the lower the weldability. A low value of $\Phi$ is beneficial since low energy is required during the welding process.

This beneficial effect is also described by heat generated during the welding process. According to Menges et al., the higher the material damping and the storage modulus, the higher the heat generation during welding and the lower the energy required for the welding process [35]. Potente [18] identifies a relationship between the heat generated during welding and the material properties, i.e., storage modulus and damping (Equation (6)).

$$q = \frac{1}{2}\left(E'\cdot\tan\delta\right)\cdot\varepsilon^2\omega \tag{6}$$

where $\varepsilon$ is the maximum strain and $\omega$ the angular frequency.

The increase in LSS follows the same trend as the loss modulus of HAR nanocomposites (Figure 7). According to Equation (6), a high loss modulus is an indicator of the effectiveness of heating the material during ultrasonic welding, effectively joining the adherents.

Nanocomposites reinforced with 0.5 wt% of G2Nan showed an improved quality of the joint compared to the unfilled polymer. The good mechanical properties of the material are conveyed on the joint capacity, maximizing both SBS and LSS.

An improvement in terms of process time could be achieved by further increasing the damping capacity and storage modulus of the ED, thanks to the low energy required and high heat generated during the welding process. The use of 2D nanofillers, such as GNPs, simultaneously improves the stiffness and damping of polymers, thanks to their high elastic modulus and high intrinsic damping capacity [36].

Although the elastic modulus of nanocomposites linearly increases with the volumetric filler fraction, experimental values are usually lower than the expected due to several

factors, such as the aspect ratio, orientation and agglomeration of nanofillers. According to shear lag theory, the transfer mechanism is mainly governed by interfacial shear stresses. In nanocomposites reinforced with 2D nanoparticles, the efficiency of reinforcement is strictly related to the nanoplatelet's geometry: a high nanoplatelet's surface-specific area promotes stress transfer, maximizing the efficiency of reinforcement. Indeed, from experimental data at 0.5 wt%, below the percolation threshold (in the absence of agglomeration phenomena) it was found that the elastic modulus of HAR nanocomposites is higher than LAR nanocomposites, meaning that the high-aspect-ratio GNPs (G2Nan) are more efficient in reinforcing the polymer compared to low-aspect-ratio GNPs (MICRO850). The effective modulus of the filler is 75 GPa for HAR and 7 GPa per LAR [37].

However, above percolation, at 2.0 wt%, the LAR nanocomposite exhibited a higher elastic modulus compared to HAR, due to the agglomeration of nanoplatelets, which impairs the efficiency of reinforcement.

Additionally, the nanofillers are usually used to improve the dissipative behavior of polymers. Similarly, the aspect ratio affects the damping properties of nanocomposites. A high aspect ratio in nanoplatelets improves the damping coefficient, promoting high energy dissipation through the large surface area [38]. However, nanofillers have never been considered as a reinforcement for coupling layers in the UW of TSCs.

Results showed a strict correlation between the LSS of welded joints and the loss modulus of the energy director. Figure 8 displays the fracture surface of LSS tests at different filler contents. In the case of the neat PVB ED, the film appears to be unmolten and there is no bond between the two adherents, resulting in the minimum LSS. When 0.5 wt% of G2Nan is added, the film is completely melted and bonded to both adherents, yielding the maximum LSS. Hence, for filler contents of 1.0 wt% and 1.5 wt%, similar failures of neat PVB can be observed. In the case of 2.0 wt% filler content, the film is completely melted, similar to the 0.5 wt% case. However, the excessive heat generation linked to the higher damping capacity of the filler leads to some thermal degradation of the substrate matrix, resulting in the exposure of the bare fibers.

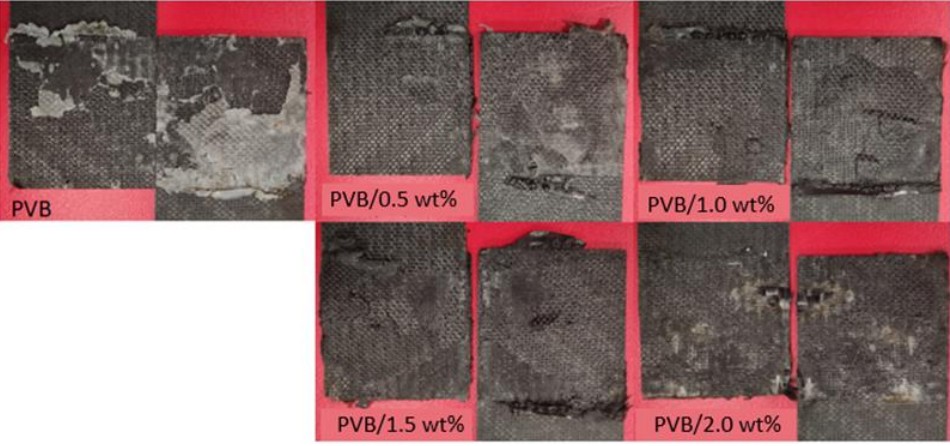

**Figure 8.** Fracture surfaces of CF/Epoxy LSS joints at different filler content.

In order to overcome the unmolten area in flat energy directors, the use of filler that improves the damping properties seems to be an effective method, also overcoming the manufacturing issues of the development of EDs with particular geometries. In fact, as stated by Villegas et al. [39], the quality of the welded joints, including the fully welded area and the lap shear strength, as well as the vibration time and the energy consumption during the welding process, were not significantly affected by the type of energy director used. However, the melting and flow behavior of the energy directors differed significantly: triangular energy directors gradually melted and flowed, while flat energy directors melted gradually but did not flow until they were completely molten.

Finally, to further optimize and increase weld strength, future investigations will be addressed. Strategies will be aimed at modifying the thickness of coupling layers and adding the PVB-reinforced film as an intermediate material co-cured with the TSC laminates. The use of GNPs/PVB coupling layers during the first stage of the co-curing process, i.e., before curing of the epoxy resin, would enable us to partially penetrate the first layer of the carbon-epoxy composite, generating macromechanical interlocking between the coupling layer and the underlying composite.

**5. Conclusions**

The possibility of using PVB/GNP coupling films as energy directors in the UW of TSC has been demonstrated. The influence of GNPs' weight content and flake aspect ratio on the mechanical properties of nanocomposites and joint quality has been assessed.

It was found that the use of lamellar nanoparticles can effectively improve the mechanical performances of polymers both in terms of damping and elastic modulus, and consequently, weldability. Unlike neat PVB, the inclusions act as reinforcement, supporting part of the load that is transferred from the matrix, both improving the elastic properties and the dissipative capabilities of the material. A correlation between SBS and storage modulus and LSS and loss modulus is found. The high hysteresis losses ensure the elevation of the temperature during welding and the high stiffness ensures the transmission of the vibrational energy from the horn to the joint interface.

An improvement of 28% of loss modulus and 80% of LSS is shown for PVB reinforced with 0.5% of G2Nan, meaning that GNPs positively affect the damping capacity of PVB and therefore the joint strength.

**Author Contributions:** Conceptualization, A.M. and B.P.; methodology, F.C. and M.G.; investigation, F.C., B.P. and L.S.; writing—original draft preparation, F.C. and B.P.; writing—review and editing, A.M. and L.S.; supervision, M.G. and A.B.; funding acquisition, A.B. and M.G. All authors have read and agreed to the published version of the manuscript.

**Funding:** This work was partially founded by Leonardo Electronics-Defence Systems BU, through the Applied Research AR01 "New materials applicable to armaments".

**Data Availability Statement:** Not applicable.

**Acknowledgments:** The authors would like to thank. Petriccione and De Tommaso for their helpful advice on technical issues and A.T.M. s.r.l. for supporting the set-up of ultrasonic welding equipment.

**Conflicts of Interest:** The authors declare no conflict of interest.

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
