# Peer review of "PVB Nanocomposites as Energy Directors in Ultrasonic Welding of Epoxy Composites"

_jcs, doi:10.3390/jcs7040160_

Round 1
Reviewer 1 Report
There are several comments on your article.
1. There is a significant decline, then rise again on figures 4,5,6 for samples with GNP content 1 wt%, (this is especially typical for the LSS vs GNP content). The explanation for this strange behavior is not given in the article. Perhaps this can be explained based on the nature of the surface morphology presented in Figure 2b.
2. The values of SBS and LSS are shown in Figures 5.6 without errors. It also raises the question why in Figure 4 for the MICRO850 samples with GNP content 0.5 wt% the errors are so small that you were not even able to depict them?
3. It is not clear why the authors decided to provide Table 2, although its contents could be conveyed in the text of the article in one sentence, that the Tg value is 72-73 C and filler content has almost no effect on temperature Tg.
Author Response
There are several comments on your article.
- There is a significant decline, then rise again on figures 4,5,6 for samples with GNP content 1 wt%, (this is especially typical for the LSS vs GNP content). The explanation for this strange behavior is not given in the article. Perhaps this can be explained based on the nature of the surface morphology presented in Figure 2b.
Thank you for this suggestion. It would have been interesting to explore this aspect. Focusing on G2nan nanocomposites, micrographs (see Figure 3) shows that at 1.0 %wt starts to agglomerate. The nanoparticles agglomeration negatively affects the mechanical properties, in fact, at 1.5 %wt the change of Young Modulus is negligible ( 3%), while loss the damping factor increases of 6%.
The energy dissipation improves with the rise of agglomeration since more energy is dissipated between adjacent nanoparticles and not only at interface between GNPs and the matrix.
This effect is beneficial for the weldability. In fact, nanocomposites damping properties can help to absorb and disperse the ultrasonic energy more efficiently, leading to a more uniform heating of the interleaved layer and lead to a stronger bond between the adherends. This effect is found in the LSS results.
The manuscript has been modified as follows, page 7 line 233: “… By increasing the filler…in energy dissipation."
The manuscript has been modified as follows, page 8 line 270: “… Even in the case … tanδ”
Eq. (5) has been added to highlight the heating phenomenon.
- The values of SBS and LSS are shown in Figures 5.6 without errors. It also raises the question why in Figure 4 for the MICRO850 samples with GNP content 0.5 wt% the errors are so small that you were not even able to depict them?
Thank you for pointing this out, error bars have been added for SBS and LSS tests. The data dispersion in the case MICRO850 samples with GNP content 0.5 wt% is very small and the error bar is depicted but it is not clearly visible.
- It is not clear why the authors decided to provide Table 2, although its contents could be conveyed in the text of the article in one sentence, that the Tg value is 72-73 C and filler content has almost no effect on temperature Tg.
The authors provided Table 2 to assess that the thermal properties of the polymer are not affected by the nanoparticles type and content. The table includes both the TGA and DSC results by describing the good agreement between the nominal filler content and the actual filler content of nanocomposites and the negligible variation of Tg. Also, the methodology employed for the determination of the actual filler content starting from TGA results has been described in the text and Eq. (3) has been added (page 5 lines 179-184)
Reviewer 2 Report
please see the attachment

Author Response
In their manuscript Cilento al. investigate properties of PVB nanocomposites. In general, the manuscript is very poorly organized. The literature review is too cursory written in a superficial way. Experimental part is too shallow. The paper lacks consistency and solid justification of the performed research. The findings are somewhat generic on the one hand and haven’t been properly analysed on the other. No comparison with previously published papers has been performed. To conclude, the manuscript suffers from lack of novelty based on the missing scientific aspects mentioned in the revision as it has no additional value for the research community. Manuscript requires significant revision to bring it to a level worthy of publication. I must therefore recommend the major revision of this paper.
Thank you for your comments. We have gone through your comments carefully and tried our best to address them one by one. The manuscript has been deeply revised in each section. We hope the manuscript has been improved accordingly.
Some of my recommendations for improvement are presented below:
- The Abstract needs significant improvement. It has a lot of missing information regarding the research results. The findings are somewhat generic. Results (in terms of numbers and their comparison) are not specified. Abstract must be more informative by providing results in terms of the numbers. Add some key values from results and highlight the novelty of this work clearly
We thank the reviewer for pointing this out. We have deeply revised the abstract addressing reviewer’s comments.
- Selected references are quite old, which from the one point of view is good, since the Authors cited necessary references to define a research problem, while from the other hand, lack of recent references may indicate an insufficiently performed literature review. I strongly recommend to add some more recent and up-to-date research papers, especially from 2021-2022; The Introduction section lacks sufficient background information. Therefore, it does not give the reader detailed background knowledge and possible wide application of this study. The literature review is very poor, in fact absent at all. The Introduction needs to be more emphasized on the research work with a detailed explanation of the whole process considering past, present and future scope. How the present study gives more accurate results than previous studies? It needs to be strengthened in terms of recent research in this area with possible research gaps. It is strongly recommended to add a recent literature. Research gaps should be highlighted more clearly and future applications of this study should be added
Thank you for this suggestion, we have exploited the literature review. The introduction section has been improved addressing reviewer’s comments.
- Before the last paragraph, the Authors are encouraged to answer the following question: What research gap did you find from the previous researchers in your field? Please discuss past studies similar or closely related to this work, mention/compare their findings, and then explain in the details how the current study brings new knowledge and difference to the field. What has been done in the course of your study. Mention it properly. It will improve the quality of the article
A discussion paragraph was added to meet the reviewer suggestions.
- In order to emphasize the importance of the subject the Authors are encouraged to discuss pultruded FRP composites that are also widely used in aerospace and automotive industries. Please, refer to: https://doi.org/10.1016/B978-0-12-819724-0.00086-0 and https://doi.org/10.1016/j.conbuildmat.2022.128694
The introduction section was deeply revised (see previous points), the suggested paper was included in the literature review.
- Suppliers of all the materials and tools used in this study should be specified 2. Section 2 gives very poor explanation of manufacturing process and testing procedures. It must be extended and the preparation process should be described in more details. I would recommend to add some pictures of manufacturing process/testing and describe it in more details. Now this part looks quite shallow. It can’t be accepted in the present form.
Section 2 was deeply revised to address the reviewer’s comments. A detailed description of manufacturing process was introduced and a schematic (figure 1) was added to reproduce each stage from plate fabrication to sample welding and testing.
- What standards did you follow for TGA, determination of thermal properties?
Thermogravimetric tests was carried out according to the ASTM E1131 - Standard Test Method for Compositional Analysis by Thermogravimetry. The manuscript was modified as follows: “Thermogravimetric analysis (TGA) (TA Instruments, Q500) was conducted to evaluate the real filler/matrix composition of the films, according to ASTM E1131.”
- Please, describe the preparation process of the specimens for morphology analysis
The modified section 2 includes a description of sample preparation.
- Units of all the parameters should be mentioned after each equation (even if dimensionless)
The manuscript was revised by including units of each parameter mentioned.
- Line 154. “Micrographs show excellent dispersion” – what do you mean by “excellent”, compared to what? How do you quantify it? Please, avoid using informal language; Line 158. “showed a very good level of homogeneity” – see my previous comment
The manuscript was modified as follow:
“Micrographs show the dispersion state in the case of MICRO850 and G2Nan. In this case of G2NAN nanocomposites (HAR), the distribution of nanoparticles is mostly uniform for all concentrations from 0.5 wt% to 2.0 wt%, with a slight level of agglomeration at high GNP content (Figure 3 b-c). Whereas MICRO850 nanocomposites (LAR) showed a better level of homogeneity with respect to G2Nan nanocomposites (HAR) and absence of clusters even at high filler content (2 wt%).”
- Lines 160-169 contain unnecessary encyclopaedical information which brings no added value to the field. Please, remove it and, in opposite, describe the results
The manuscript was modified as follows:
“By increasing the filler percentage, the HAR form local clusters since there is an abrupt increase of viscosity which threaten the mixing quality. As result, a reduction in stiffness and damping was observed, likely related to poor interface between the GNPs and the matrix and to the presence of agglomeration particles.
At low particle content there was a concurrent increase in mechanical performance and dissipative behaviour while the poor dispersion at higher content worsens the stress transfer resulting in a reduction of storage modulus. In addition, internal slippage between particles leads to a progressive increase in energy dissipation [25]”.
- Section 5 follows Section 3. Section 4 is missing
Fixed
- All in all, Sections 2, 3 and 5 look very poor to me. This is not the level of the proper scientific article
The manuscript has been deeply revised to address the reviewer’s comments.
- The results are merely described and are limited to comparing the experimental observation. The Authors are encouraged to include Discussion Section and critically discuss the observations from this investigation with existing literature; The Authors are encouraged to bring additional paragraph discussing possibilities for future studies, showing future research directions to the scholars. Besides, deeper analysis regarding the possible practical applications of results revealed in this study is needed
A discussion paragraph was added to meet the reviewer suggestions.
Round 2
Reviewer 2 Report
All major comments were adequately addressed and the Authors have done an admirable job of improving the quality of the manuscript. Therefore, it can be accepted without any structural modification.